# Immunomodulatory Effects of Non-Thermal Plasma in a Model for Latent HIV-1 Infection: Implications for an HIV-1-Specific Immunotherapy

**DOI:** 10.3390/biomedicines11010122

**Published:** 2023-01-03

**Authors:** Hager Mohamed, Rachel Berman, Jennifer Connors, Elias K. Haddad, Vandana Miller, Michael R. Nonnemacher, Will Dampier, Brian Wigdahl, Fred C. Krebs

**Affiliations:** 1Department of Microbiology and Immunology, Drexel University College of Medicine, Philadelphia, PA 19102, USA; 2Center for Molecular Virology and Gene Therapy, Institute for Molecular Medicine & Infectious Disease, Drexel University College of Medicine, Philadelphia, PA 19102, USA; 3Division of Infectious Diseases & HIV Medicine, Department of Medicine, Drexel University College of Medicine, Philadelphia, PA 19102, USA

**Keywords:** immunomodulation, people living with HIV (PLWH), HIV-1, latency, reactivation, non-thermal plasma (NTP), low temperature plasma (LTP), chemotaxis

## Abstract

In people living with HIV-1 (PLWH), antiretroviral therapy (ART) eventually becomes necessary to suppress the emergence of human immunodeficiency virus type 1 (HIV-1) replication from latent reservoirs because HIV-1-specific immune responses in PLWH are suboptimal. Immunotherapies that enhance anti-HIV-1 immune responses for better control of virus reemergence from latent reservoirs are postulated to offer ART-free control of HIV-1. Toward the goal of developing an HIV-1-specific immunotherapy based on non-thermal plasma (NTP), the early immunological responses to NTP-exposed latently infected T lymphocytes were examined. Application of NTP to the J-Lat T-lymphocyte cell line (clones 10.6 and 15.4) stimulated monocyte recruitment and macrophage maturation, which are key steps in initiation of an immune response. In contrast, CD8+ T lymphocytes in a mixed lymphocyte reaction assay were not stimulated by the presence of NTP-exposed J-Lat cells. Furthermore, co-culture of NTP-exposed J-Lat cells with mature phagocytes did not modulate their antigen presentation to primary CD8+ T lymphocytes (cross-presentation). However, reactivation from latency was stimulated in a clone-specific manner by NTP. Overall, these studies, which demonstrated that ex vivo application of NTP to latently infected lymphocytes can stimulate key immune cell responses, advance the development of an NTP-based immunotherapy that will provide ART-free control of HIV-1 reactivation in PLWH.

## 1. Introduction

Over 38 million people worldwide are currently infected with the human immunodeficiency virus type 1 (HIV-1) [1]. Antiretroviral therapy (ART) effectively suppresses productive HIV-1 replication in people living with HIV (PLWH), but does not eliminate or reduce HIV-1 reservoirs comprised of latently infected cells that harbor proviral DNA integrated into their host cell genomes. Although the provirus in a latently infected cell can remain transcriptionally silent for years [2,3], it can be reactivated. This can occur if ART treatment is ineffective or halted, resulting in productive HIV-1 replication, reemergence of the virus, an attendant rebound in viremia, and progression of clinical disease. Higher levels of virus replication result in the rapid loss of CD4+ T lymphocytes, facilitate the emergence of drug-resistant HIV-1 quasispecies, and promote progression toward late-stage disease and acquired immunodeficiency syndrome (AIDS) [2]. Because ART does not eliminate latent reservoirs [4], PLWH must maintain lifelong adherence to ART to keep HIV-1 production from latent reservoirs in check [5,6,7].

The reemergence of HIV-1 from latently infected reservoir cells is also facilitated by suboptimal host innate and adaptive immune responses. Several defects in the innate and adaptive immune responses against HIV-1-infected cells contribute to consistently poor immune suppression of recrudescent infection when ART fails or is halted. Antigen-presenting cells (APCs) are reported to be inefficient in recognition and uptake of infected cells, and they have dysregulated pro-inflammatory cytokine and chemokine release, disrupted maturation, and impaired cross-priming ability [8,9]. Since APCs are necessary to prime CD8+ T lymphocytes, the compromised APC responses likely contribute to the suboptimal CD8+ T-lymphocyte response, resulting in the inefficient killing of infected cells often reported in HIV-1-infected individuals.

An immunotherapy that resolves or offsets these immune deficiencies and facilitates ART-free control of HIV-1 reemergence would be an enormous benefit to PLWH, who are currently committed to lifelong regimens of ART [10]. We are investigating the use of non-thermal plasma (NTP) as the basis for an immunotherapy that provides ART-free control of HIV-1 recrudescent infection. NTP, which is also called low temperature plasma (LTP), cold atmospheric plasma (CAP), and gas plasma, is being studied as the basis for multiple applications in the expanding field of plasma medicine, including treatments for cancer and a means to facilitate wound healing and disinfection [11]. NTP is a partially ionized gas composed of electrons, ions, charged particles, and a predominance of reactive oxygen and nitrogen species (RONS). The generation of NTP also exposes cells to local electric fields and small amounts of ultraviolet radiation. Current dogma attributes most of the biological effects of NTP to oxidative stress imposed on cells by the RONS that trigger endoplasmic stress response pathways [12,13]. The NTP-imposed oxidative stress boosts cellular immunogenicity by inducing the emission of damage-associated molecular patterns (DAMPs) that stimulate several immune functions, including APC recruitment, phagocytosis, and maturation [14,15]. The ability of NTP to promote APC responses and subsequent adaptive immune responses has previously been demonstrated in murine models of anti-tumor NTP therapy [14,15,16,17,18,19,20,21].

In the proposed immunotherapy for HIV-1 for PLWH, the patient’s own cells, exposed ex vivo to NTP, will be used as an immunogen. As a first step toward the development of this immunotherapy, we previously demonstrated that NTP enhances the immunogenicity of HIV-1 latently infected cells using the J-Lat (clone 10.6) CD4+ T-lymphocyte cell line [22]. This cell line, which has been used extensively as a model for latent HIV-1 infection, was established from the parent Jurkat T-lymphocyte cell line through integration of an HIV-1 provirus that lacks the genes for Env and Nef and contains the GFP gene as a reporter for viral gene expression. Clones of the J-Lat cell lineage contain a transcriptionally silent provirus that can be reactivated using latency reversal agents (LRA) or stimulating agents, such as phorbol 12-myristate 13-acetate (PMA) [23]. In our previous study, we demonstrated that NTP exposure stimulated the emission of immunogenic DAMPs that promote immunologically important functions of APCs [22]. NTP application to J-Lat cells also caused reactivation of viral gene expression. Furthermore, while NTP-mediated emission of DAMPs was observed in both J-Lat cells and the parent Jurkat cell line (which lacks an HIV-1 provirus), J-Lat cells emitted DAMPs more robustly, possibly as a result of additive effects of viral gene expression and NTP-induced immunogenicity [22]. Our observations suggest that NTP may be used to boost the immunogenicity of latently infected cells and to drive anti-HIV-1 immunity, which is the basis of our proposed immunotherapy approach for HIV-1 infection.

In the present study, we examined the effects of the proposed NTP-modified immunogen on other cell types involved in mounting an immune response at the site of administration. We evaluated changes in APC and CD8+ T-lymphocyte functions after co-culture with DAMP-emitting NTP-exposed J-Lat cells. As a first step toward addressing the potential impact of intra-patient reservoir cell diversity on NTP-stimulated immunogenicity, we also examined the effects of NTP on viral gene expression in two J-Lat cell clones (10.6 and 15.4) that differ in the provirus integration sites [23]. We report for the first time potential immunomodulatory mechanisms of NTP that can be harnessed for eliciting ART-free control of HIV-1 recrudescent infection.

## 2. Materials and Methods

### 2.1. Cell Culture

J-Lat CD4+ T lymphocytes (HIV Reagent Program: clone 10.6, ARP-9849; clone 15.4, ARP-9850) and THP-1 cells (ATCC^®^TIB-202™) were cultured in RPMI-1640 supplemented with L-Glutamine, 10% fetal bovine serum (FBS), and 1% penicillin/streptomycin. Purified primary CD8+ T lymphocytes were purchased as de-identified cell preparations from the University of Pennsylvania Human Immunology Core (Philadelphia, PA, USA). Because the cells were purchased as de-identified cell preparations, experiments did not involve any donor recruitment, consenting, sample collection, or sample preparation, and were exempt from human subjects considerations. Cell cultures containing CD8+ T lymphocytes were supplemented with recombinant IL-2 (Fisher Scientific, Waltham, MA, USA). All cells were maintained and incubated at 37 °C with 5% CO_2_ and 95% humidity for all experiments.

### 2.2. Non-Thermal Plasma Application

J-Lat cells (50 µL at 4 × 10^6^ cells/mL) were aliquoted into each well of a 24-well plate to seed 200,000 cells per well. NTP was generated using a nanosecond pulsed dielectric barrier discharge (nspDBD) device. The nspDBD device was operated at an applied voltage of 29 kV, a pulse width of 20 ns, and a fixed NTP exposure time of 10 s. The dose of NTP was varied by changing the pulse frequency (30 or 150 Hz) and was delivered to cells using a 24-well-sized electrode positioned 2 mm above the bottom of the well. Following NTP exposure of cell aliquots, cells in each well were immediately supplemented with 450 µL of RPMI for a final cell density of 4 × 10^5^ cells/mL in each well. Cells were then incubated for 24 h prior to assays.

### 2.3. Migration Assay

J-Lat cells were seeded into 24-well plates and exposed to NTP (30 or 150 Hz) or treated with 250 nM mitoxantrone (MTX). Immediately following exposure, transwell inserts (3 µm pore size) were positioned in each well (*n* = 3 for each condition) and loaded with THP-1 monocytes pre-labeled with 0.5 µM CellTrace Far Red (CTFR) (ThermoFisher Scientific, Waltham, MA, USA, Cat# C34564). The number of CTFR-positive THP-1 monocytes that migrated into the basolateral chamber in each well was measured 24 h later using a BD LSRFortessa™ flow cytometer.

### 2.4. Phagocytosis and Maturation Assay

THP-1 monocytes were differentiated into M0 macrophages by treatment with 100 nM PMA for 4 days. A phagocytosis assay was then performed as previously described [22], with M0 and J-Lat 10.6 or 15.4 cells co-cultured for 24 h to allow maximal phagocytosis. M0 macrophages were then washed twice with PBS to remove non-internalized J-Lat cells and incubated for 24 h to allow antigen processing. Following incubation, macrophages were washed with PBS, detached, and incubated in blocking buffer (2% FBS in PBS) containing TruStain Fc Receptor Blocker (BioLegend, San Diego, CA, USA, Cat# 422301). Macrophages were then stained with Alexa Fluor^®^ 594 anti-MHC I (Novus Biologicals, Centennial, CO, USA, Cat# FAB7098T), PE anti-HLA-DR (BioLegend, Cat#307605), FITC anti-CD80 (Biolegend, Cat#305205), and Brilliant Violet 785 anti-CD86 (BioLegend, Cat#305441) at 4 °C. Maturation maker expression was measured using a BD LSRFortessa™ flow cytometer. Data were analyzed using FlowJo™ Software (Becton, Dickinson, & Company, Ashland, OR, USA) after doublet exclusion using forward scatter (FSC-A vs. FSC-H).

### 2.5. Cross-Presentation Assay

THP-1 monocytes were seeded at 1 × 10^5^ cells per well in a 24-well plate and stimulated with 100 nM PMA for 4 days to induce differentiation into mature macrophages (M0). J-Lat cells were exposed to 150 Hz NTP, then cultured for 24 h, as described previously. Cells were then washed with PBS and co-cultured with M0 macrophages at a 1:1 ratio for 24 h to allow antigen processing and peptide display. After the 24 h incubation, target macrophages were washed and fixed with 0.5% PFA prior to co-culturing with CTFR-labeled CD8+ T lymphocytes at E:T ratios of 1:1, 5:1, and 10:1. Co-cultures were supplemented with IL-2 and incubated for 1 and 3 days. CD8+ T lymphocytes treated with an anti-CD3 antibody (Clone OKT3) (BioLegend, Cat#317302) and an anti-CD28 antibody (Clone CD28.2) (BioLegend, Cat#302902) cocktail or PMA and Ionomycin served as positive controls for proliferation and activation. Following incubation, CD8+ T lymphocytes were washed with PBS and stained with PE anti-CD8 (BioLegend, Cat#344705), FITC anti-human CD25 (BioLegend, Cat#356105), and PE/Dazzle™ 594 anti-human CD69 (BioLegend Cat#310941) antibodies on ice, then stained with LIVE/DEAD™ Fixable Aqua Dead Cell Stain (ThermoFisher Scientific, Cat# L34957) to label dead cells. Expression of CD69, CD25, and CD8+ T-lymphocyte proliferation was measured using a BD LSRFortessa™ flow cytometer. Data were analyzed using FlowJo™ Software after doublet exclusion using forward scatter (FSC-A vs. FSC-H) and gating on live CD8+ T lymphocytes.

### 2.6. Mixed Lymphocyte Reaction (MLR)

J-Lat cells were exposed to NTP, then cultured for 24 h, as described previously. Prior to co-culture, J-Lat cells were washed with PBS and treated with 0.5% PFA or 10 µg/mL Mitomycin C (Sigma-Aldrich, St. Louis, MO, USA, Cat#M4287) for 2 h at 37 °C to inhibit proliferation of J-Lat cells, then washed 3× with PBS and supplemented with RPMI. Target J-Lat cells were then co-cultured with Cell Trace Far Red (CTFR) pre-labeled CD8+ T lymphocytes at effector-to-target (E:T) ratios of 1:1, 5:1, and 10:1. CD8+ T lymphocytes treated with a cocktail containing 2 µg/mL of an anti-CD3 antibody (Clone OKT3) (BioLegend, Cat#317302) and 2 µg/mL of an anti-CD28 antibody (Clone CD28.2) (BioLegend, Cat#302902) or 10 ng/mL PMA and 400 ng/mL Ionomycin served as positive controls for proliferation. Co-cultures were supplemented with 10 ng/mL IL-2 incubated for 1 and 7 days. Following incubation, cells were washed with FACS Buffer and stained with PE anti-CD8 antibody (BioLegend, Cat#344705), then stained with LIVE/DEAD™ Fixable Aqua Dead Cell Stain (ThermoFisher Scientific, Cat# L34957) to label dead cells. The percent of proliferated CD8+ T lymphocytes in co-cultures was enumerated using a BD LSRFortessa™ flow cytometer. Data were analyzed using FlowJo™ Software after doublet exclusion using forward scatter (FSC-A vs. FSC-H), and gating on live CD8+ T lymphocytes with a low intensity of CTFR.

### 2.7. Analysis of the Chromatin State at LTR Integration Sites

The Epilogos browser [24,25] was used to visualize the chromatin states at the integration sites in J-Lat clones 10.6 and 15.4. The chromatin state was analyzed in a 5 kb window around the integration sites of clone 10.6 in an intron of the SEC16A gene and of clone 15.4 in an intron of the UBA2 gene, as well as at a single base resolution at each of the sites. The transcription state was characterized as ‘strong’ or ‘weak.’ Transcription patterns in the following primary blood and T cell samples compiled for the Epigenome Roadmap consortium were used to analyze the chromatin states at the integration sites: T lymphocytes from cord blood (E033), T lymphocytes from peripheral blood (E034), T helper cells from peripheral blood (E037, E040), T helper naive cells from peripheral blood (E038, E039), T helper cells PMA-I stimulated (E041), T helper 17 cells\PMA-I stimulated (E042), T helper cells from peripheral blood (E043), T regulatory cells from peripheral blood (E044), Effector/memory T lymphocytes enriched from peripheral blood (E045), CD8+ naïve cells from peripheral blood (E047), CD8+ memory cells from peripheral blood (E048), and mononuclear cells from peripheral blood (E062).

### 2.8. Statistical Analyses

Data are representative of at least three independent experiments (*n* ≥ 3 each), unless otherwise stated. Graphing and statistical analyses were performed using Prism 9 (GraphPad Software, La Jolla, CA, USA). Mean and standard errors were calculated, and statistical significance was determined using the Kruskal–Wallis test with Dunnett’s post-hoc test for the migration and HIV-1 reactivation assays, a Brown–Forsythe and Welch’s Anova for the maturation assay, and a one-way Anova with Dunnett’s post-hoc test for the cross-presentation assay and MLR.

## 3. Results

Experiments were conducted to demonstrate the effects of NTP on latently infected T lymphocytes, as well as the effects of NTP-induced changes in T lymphocytes on antigen-presenting cells (APCs). The J-Lat and THP-1 cells were chosen as models of these respective cell types.

### 3.1. Effect of NTP on Migration of Monocytes toward NTP-Exposed J-Lat Clones

An effective immune response to the proposed immunotherapy necessitates recruitment of immune cells toward NTP-exposed, HIV-1-infected cells at the site of administration. The initiation of a robust innate immune response is critical to mounting adaptive, systemic, anti-HIV-1 immunity.

We previously demonstrated that NTP-exposed cells release several chemotactic DAMPs, such as IL-1β, CCL2 (MCP-1), and HMGB1 [22]. These act as “find me” molecules for APCs and serve to recruit them toward DAMP-emitting cells [15,26,27,28,29,30]. To demonstrate promotion of chemotaxis by NTP, we used an in vitro transwell system and assayed monocyte migration toward NTP-exposed J-Lat cells (clone 10.6). J-Lat cells were exposed to nanosecond-pulsed dielectric barrier discharge (DBD) plasma (29.5 kV at 30 or 150 Hz for 10 s) and then incubated for 24 h in the basolateral chamber of a transwell system to allow for chemotactic DAMP emission, as previously observed [22]. Immediately following the NTP exposure of J-Lat cells, transwell inserts were placed into wells containing J-Lat cells, and Cell Trace Far Red (CTFR)-labeled THP-1 monocytes (APCs) were placed in the apical chamber separated from the J-Lat cells by a semi-permeable membrane. In this setup, DAMPs produced by J-Lat cells should cause the THP-1 monocytes to move from the apical chamber into the basolateral chamber. The migration of THP-1 monocytes was evaluated by quantification of CTFR-labeled cells in the basolateral chamber after an additional 24 h in transwell co-culture with NTP-exposed cells (Figure 1a). These THP-1 monocytes were quantified using flow cytometry, by first excluding debris and doublets, then gating on the CTFR high population (Figure 1b). A 1.96-fold increase was observed in the number of THP-1 monocytes that migrated toward J-Lat cells exposed to DBD plasma generated at 150 Hz relative to migration toward NTP-naïve J-Lat cell controls (Figure 1b). In an apparent dose response to NTP application, NTP generated at 30 Hz was insufficient to promote THP-1 migration. This may be partially due to an NTP dose-dependent increase in cell death. In previous studies [31], the majority of cells exposed to 30 Hz NTP cells remained viable, while cells exposed to NTP generated at 150 Hz were less than 70% viable. To verify that NTP-induced chemotaxis was not unique to the 10.6 J-Lat clone, the migration of THP-1 cells in transwell co-culture with cells of the J-Lat 15.4 clone exposed to NTP (150 Hz) was evaluated. This clone contains the same provirus as clone 10.6 integrated at a different location in the host cell genome. Differences in integration site have been previously shown to affect HIV-1 gene expression and the immunogenic potential of each clone [32,33,34]. THP-1 migration to the NTP-exposed J-Lat clone 15.4 was comparable to migration toward clone 10.6 (Figure 1b), suggesting that the site of provirus integration (at least these two sites) does not affect the magnitude of the APC migration stimulated by the production of NTP-induced chemotactic DAMPs.

### 3.2. Effect of NTP on Macrophage Maturation Markers Following Uptake of NTP-Exposed Cells

The previous experiments demonstrated that NTP exposure of latently infected T lymphocytes stimulates chemotaxis of monocytes. In vivo, these recruited monocytes would differentiate or mature into APCs (macrophage or dendritic cells) and phagocytose the DAMP-emitting cells. We have previously demonstrated DAMP-emitting J-Lat cells are phagocytosed by THP-1 M0 macrophages [22]. Following phagocytosis, maturation of APCs is critical for engagement with adaptive immune cells and is a key step for priming the T-lymphocyte response [35]. Maturation is indicated by elevation of co-stimulatory molecules, along with the antigen-presenting molecules MHC I or HLA-DR (MHC II) [27,35]. To assay APC maturation in our in vitro system, THP-1 macrophages were cultured with J-Lat cells (clone 10.6 or 15.4) that had been exposed 24 h prior to NTP (150 Hz). Non-internalized J-Lat cells were washed off, and the THP-1 macrophages were then labeled with antibodies against MHC I, HLA-DR, and the costimulatory molecules CD80 and CD86. Flow cytometry was used to measure the mean fluorescence intensities (MFI) of the maturation marker antibodies on THP-1 macrophages. Engulfment of NTP-naïve J-Lat cells (untreated) resulted in slight increases in MHC I, but not HLA-DR (Figure 2), indicating a baseline level of maturation, likely attributable to uptake of genetically heterologous J-Lat cells by THP-1 cells. Engulfment of NTP-exposed clone 10.6 cells resulted in a 1.33-fold increase in surface expression of MHC I and a 1.13-fold increase in HLA-DR on THP-1 macrophages 24 h post co-culture with the J-Lat cells relative to marker levels on immature M0 macrophages cultured alone (Figure 2). Engulfment of NTP-exposed cells of clone 15.4 caused no increase in MHC I, but a slight elevation (1.10-fold) of HLA-DR. These small, but statistically significant changes are indicative of APC maturation subsequent to engulfment of cells exposed to NTP.

Another indicator of macrophage maturation is the upregulation of co-stimulatory molecules, such as CD80 or CD86. Previous studies showed that application of NTP to cancer cells stimulated APC maturation, as evidenced by increased surface expression of the co-stimulatory molecule CD86 [19,36]. Therefore, the expression of CD86 and CD80 on macrophages following their phagocytosis of NTP-exposed J-Lat cells (clone 10.6 or 15.4) was measured. CD80 was significantly upregulated after phagocytosis of J-Lat clone 15.4 (1.23-fold relative to CD80 expression on NTP-naïve M0 macrophages), but not after uptake of clone 10.6 (Figure 2). CD86 expression was increased 2.2-fold only on macrophages that had engulfed NTP-exposed cells of clone 10.6. These results demonstrate that NTP exposure of HIV-1 latently infected cells can boost APC maturation (Figure 2).

### 3.3. Effect of NTP-Exposed J-Lat Cells on Antigen Cross-Presentation by Macrophages

In HIV-1-infected individuals, defective innate immune responses cause inefficient priming of cells of the adaptive immune system, which contributes to suboptimal systemic targeting of infected cells. We demonstrated in our previous and present studies that NTP can boost monocyte recruitment, phagocytosis of latently infected cells by macrophages, and the maturation of macrophages. Once these responses occur in vivo following therapeutic vaccination with NTP-exposed cells, the next event in the stimulation of an adaptive immune response is the presentation of antigen(s) by the APCs that migrated into the lymph node to resident T lymphocytes in the lymph node. Antigens displayed by APCs from the phagocytosis of virus-infected cells may be cross-presented on MHC I to CD8+ T lymphocytes in the lymph node. This in vivo cross-presentation, or cross-priming, initiates a CD8+ T-lymphocyte response that will target infected cells throughout the body [37,38].

Because J-Lat clone-dependent increases in MHC I and co-stimulatory molecule cell surface display were observed in response to NTP-exposed J-Lat cell engulfment (Figure 2), we hypothesized that these effects would contribute to NTP induction of cross-presentation. To test this hypothesis, the cross-presentation of antigens to CD8+ T lymphocytes in an in vitro co-culture system was assessed.

THP-1-derived macrophages that had engulfed either NTP-exposed J-Lat cells (THP-1-JL-150 Hz) or NTP-naïve J-Lat cells (THP-1-JL-Untreated) were each co-cultured with primary CD8+ T lymphocytes from healthy donors. Varied effector:target (E:T) ratios of CD8+ T lymphocytes (effectors) to THP-1 macrophages (targets) were used to measure the extent of CD8+ T-lymphocyte stimulation. CD8+ T-lymphocyte cross-priming was assessed via flow cytometry, through the expression of activation markers CD69 and CD25 on viable CD8+ T lymphocytes at one- and three-days post co-culture with the THP-1 macrophages. While an increase in CD25 display was observed on CD8+ T lymphocytes co-cultured with macrophages that had engulfed J-Lat cells, there was no statistically significant difference between CD8+ T lymphocytes co-cultured with THP-1 cells alone, THP-JL-Untreated cells, or THP-JL-150 Hz cells at any E:T ratio (Figure 3a,b). Furthermore, levels of CD69 and CD25 did not appear to differ between engulfed J-Lat clones 10.6 and 15.4. As verification of their responsiveness to stimulation, CD8+ T lymphocytes robustly displayed increased levels of CD69 and CD25 (primarily on day 3) in response to stimulation by a PMA and ionomycin (PMA + Iono) cocktail. These results suggest that uptake of NTP-exposed J-Lat cells on its own is not sufficient to stimulate THP-1 macrophages to efficiently cross-present viral antigens in vitro in our system.

### 3.4. Effect of NTP on Antigenicity of J-Lat Cells

We previously demonstrated that NTP exposure of J-Lat clone 10.6 cells increased surface MHC I expression and stimulated the display of novel peptides on MHC I, thus potentially enhancing the antigenicity of these cells [22]. Because of this increased antigenicity, inoculated cells in our proposed ex vivo immunotherapy approach may be subject to increased recognition and targeting by local CD8+ T lymphocytes. Local or tissue-resident CD8+ T lymphocytes serve to eradicate infected cells by inducing cytolysis and are found to be more effective in elite controllers, a minority of PLWH that control HIV-1 reemergence from latent reservoirs in the absence of ART [9,39]. We hypothesized that NTP exposure of J-Lat cells stimulates targeting by CD8+ T lymphocytes through increasing antigenic peptide presentation and subsequent CD8+ T-lymphocyte proliferation. This hypothesis was tested using a mixed lymphocyte reaction (MLR), with NTP-exposed J-Lat cells serving as target cells and responding primary CTFR-labeled CD8+ T lymphocytes as the effectors. In these experiments, little or no CD8+ T-lymphocyte proliferation was observed in response to co-culture with NTP-exposed J-Lat cells (Figure 4). In contrast, stimulation of CD8+ T lymphocytes with PMA and ionomycin resulted in up to a 55.81-fold increase in the percentage of proliferated CD8+ T lymphocytes relative to unstimulated CD8+ T lymphocytes seven days post-treatment (Figure 4). Stimulation with anti-CD3 and anti-CD28 antibodies resulted in lower levels of stimulation (relative to PMA + ionomycin), with somewhat higher donor-dependent variability. These results suggest NTP-induced increases in cell surface MHC I and NTP-associated changes in peptide display are not associated with increased antigenicity, at least within the context of this experimental design.

### 3.5. Effect of NTP on HIV-1 Gene Expression in J-Lat Clones 10.6 and 15.4

In experiments that examined the effect of J-Lat phagocytosis on macrophage maturation (Figure 2), cell surface MHC I and HLA-DR upregulation varied between the 10.6 and 15.4 J-Lat clones. This clone-dependent effect may be attributed to differences in the sites of provirus integration into the host genome and the resulting differences in viral gene expression, as driven by transcription regulated by the long terminal repeat (LTR). We previously demonstrated that NTP induces HIV-1 reactivation in J-Lat clone 10.6, albeit at modest levels. The increase in viral gene expression was also associated with greater NTP-induced immunogenicity (relative to that of the Jurkat parent cell line, which does not have an integrated provirus) [22], suggesting that differences in the immunostimulatory effects of NTP between J-Lat clones 10.6 and 15.4 may be due to differences in reactivated HIV-1 gene expression induced by NTP exposure. To evaluate this, we assayed GFP expression in both J-Lat clones 24 h after NTP exposure. NTP application to J-Lat clone 15.4 cells resulted in considerably higher levels of GFP expression (19.52-fold) than levels of GFP in NTP-exposed cells of J-Lat clone 10.6 (3.79-fold over NTP-naïve cells) (Figure 5). In contrast, stimulation with 100 nM PMA (positive control) stimulated higher viral gene expression in clone 10.6 cells (42.89-fold) compared to clone 15.4 cells (4.89-fold). These results indicate that the level of HIV-1 reactivation from latency is dependent on the provirus integration site. As PMA is an agonist of protein kinase C (PKC) [40], our data also suggest that NTP stimulates LTR activity through pathways that are distinct from those involving PKC.

To investigate the underlying cause of differences in NTP-mediated latency reversal between clone 10.6 and 15.4, we considered the influences of the distinct genomic landscapes surrounding the proviruses in each clone. Stimulation of HIV-1 gene expression can be heavily influenced by the site of the HIV-1 provirus integration, as well as the local chromatin architecture, which would either facilitate or hinder transcription, depending on the accessibility of the DNA [3,32,41,42]. Furthermore, the integration sites in latently infected cells vary between HIV-1 patients and are key determinants in the reactivation efficacies of LRAs, which are under study as the “shock” agent in “shock and kill” HIV-1 eradication strategies [43]. This, in turn, influences the effectiveness by which immune cells will recognize and target latently infected populations across individuals [44,45]. This variability is reflected in the experimental J-Lat cell line models for latent infection, where clones containing distinct integration sites for the HIV-1 provirus respond with different levels of virus reactivation to the same stimulant, such as PMA or TNF-α [23,34,46].

We speculated that NTP-induced viral gene expression was being modulated differentially by the chromatin structures surrounding the integration sites in clones 15.4 and 10.6. The influence of chromatin structure on NTP-stimulated gene expression is suggested by prior reports of NTP-mediated modulation of chromatin architecture [47,48]. To gain insights into the enhanced NTP-induced HIV-1 reactivation in clone 15.4, the Epilogos browser was used to examine the chromatin state at each integration site. The Epilogos browser [25] provides visual representations—epilogos—of chromatin states identified in human primary blood and T-lymphocyte samples from the Epigenome Roadmap consortium [24]. Both clones of the J-Lat cells used in this study contain a single copy of a provirus integrated into intronic regions of human genes. Analyses of the integration sites located in the SEC16A and UBA2 genes in clones 10.6 and 15.4, respectively, revealed a higher fold enrichment score for transcriptional activity in clone 10.6 than in clone 15.4 (Appendix A). This is consistent with our observation that PMA stimulation induced more HIV-1 gene expression in clone 10.6 than in 15.4 (Figure 5). However, this finding is counter to our observation that viral gene expression was higher in clone 15.4 in response to NTP application and suggests that NTP-mediated latency reversal occurs through different pathway(s) relative to PMA. The responses of the two J-Lat clones to NTP are consistent with the variable reactivation responses of these cells to other LRAs. These results suggest that following NTP application to peripheral blood cells from PLWH, variable responses to NTP-induced reactivation and associated virus production can also be expected due to variations in integration sites between patients. Nevertheless, levels of stimulated viral gene expression may still be sufficient to raise the HIV-1 antigenicity and immunogenicity of NTP-exposed cells in the ex vivo portion of the proposed immunotherapy, especially considering the increased immunogenicity of these cells.

## 4. Discussion

Antiretroviral drugs are effective means for suppressing HIV-1 replication in PLWH in the absence of an effective host CD8+ T-lymphocyte response. However, the requirement for lifelong adherence of ART, coupled with the toxicity of the drugs, the cost of therapy, the need for compliance with potentially complex treatment regimens, and the emergence of resistant HIV-1 strains, make adherence to ART challenging for millions of infected individuals [1]. There is a dire need for simple, sustainable, and cost-effective interventions for HIV-1 infection and associated disease that provide at least a functional cure, if not eradication of the virus. In this study, the immunomodulatory potential of NTP was explored as a step toward the development of an NTP-based immunotherapy for HIV-1 recrudescent infection.

The most actively explored use of NTP technology is for the treatment of another persistent disease: cancer. Besides its direct and selective cytotoxicity against cancer cells, NTP has also been shown to strengthen immunity by enhancing multiple functions of cells involved in innate and adaptive immune responses [13,19,49]. However, the use of NTP to overcome immune defects associated with infectious diseases has only recently garnered attention. We have previously shown that application of NTP to cells latently infected with HIV-1 increases their immunogenicity, as evidenced by the emission of DAMPs after NTP exposure [22]. The current study focused on characterizing the effect of NTP-exposed HIV-1-infected cells on activities of naïve APCs, specifically migration, phagocytosis, and maturation. These will be important in vivo responses to inoculated NTP-exposed cells intended to stimulate APCs and the subsequent adaptive immune response. We further examined the effect of NTP-exposed HIV-1-infected cells on the function of CD8+ T lymphocytes, which are responsible for adaptive immune control of HIV-1.

These studies are part of the development of an immunotherapy strategy that will offer a durable control of viral replication through resolution of or compensation for defects in the anti-HIV-1 immune response. In the envisioned immunotherapy approach, patient leukocytes (which will include small numbers of latently infected CD4+ T lymphocytes) will be collected and exposed ex vivo to NTP to induce pro-phagocytic and chemotactic DAMP emission. These DAMP-emitting cells will then be administered to the patient as an autologous therapeutic vaccine. This therapeutic approach is hypothesized to enhance HIV-1-specific immune responses by boosting innate and adaptive immune responses at the site of administration through enhanced immunogenicity and adjuvanticity. Based on our previous observation that NTP alters the display of MHC class I-associated peptides on latently infected cells [22], a broader adaptive HIV-1-specific immune response through epitope spreading is also anticipated. An additional strength of this approach is that the treatment will be personalized because the antiviral immune responses will be mounted against the patient’s own viral genotypes.

Two important roles for NTP in our proposed immunotherapy are enhancing APC recruitment and increasing APC phagocytic capacity. Deficits in these activities in PLWH compromise the development of protective immune responses [50]. We have previously shown that NTP increases the phagocytic uptake of NTP-exposed J-Lat cells [22]. Prior to phagocytosis, APCs must be conscripted to the site. Here, we showed that chemotaxis of monocytes was stimulated when cultured with either of the two clones of J-Lat cells (Figure 1). This critical step in the initiation of innate immune response enables early recognition and phagocytosis of HIV-1-infected cells. Our data are consistent with in vivo studies demonstrating that various innate cell populations infiltrate NTP-exposed tumors [15,51]. In our immunotherapy approach, we expect that introduction of the patient’s cells after NTP exposure will recruit local APCs (e.g., Langerhans cells) to the area of inoculation, which will take up the inoculated cells by phagocytosis and initiate an immune response.

After recruitment to the site of inoculation and active phagocytosis, APCs undergo functional maturation, as indicated by upregulation of specific cell surface markers. Our studies showed that the uptake of NTP-exposed cells induced macrophage maturation. However, the maturation profile induced by each J-Lat clone was distinct (Figure 2). As our maturation assay included an overnight co-culture period, it is reasonable to assume that factors other than phagocytosis may have influenced APC maturation. Specifically, macrophage maturation may have also been modulated by the clone-specific display or release of pathogen-associated molecular patterns (PAMPs) from NTP-exposed J-Lat cells, as well as the release of pro-inflammatory cytokines during the incubation period. Additionally, we found that macrophages responding to NTP-exposed cells release high concentrations of IL-1β, as well as other pro-inflammatory cytokines known to promote Th1 responses necessary for antiviral immunity [22]. These are additional immunomodulatory mechanisms of NTP-exposed cells that have yet to be investigated in the context of infection by HIV-1 (and other viruses).

Following efficient recruitment, phagocytosis, and maturation, the subsequent migration of APCs to the lymph node and presentation of infected cell-derived antigens to CD8+ T lymphocytes is necessary to build anti-HIV-1 immunity. We assayed for this APC function using an in vitro cross-presentation assay, in which no increase in activated primary CD8+ T lymphocytes responding to THP-1 macrophages that had engulfed NTP-exposed J-Lat cells (relative to THP-1 macrophages that had engulfed untreated J-Lat cells) was observed. Mass spectrometry analysis will be necessary to verify that presented peptides are indeed derived from HIV-1 proteins expressed in the engulfed J-Lat cells, as opposed to other J-Lat cell or THP-1-derived peptides. Furthermore, efficient cross-presentation requires not only presentation of antigenic peptides to CD8+ T lymphocytes and binding to co-stimulatory molecules, but also the appropriate pro-inflammatory cytokine signals emitted by the APCs during the interaction [52]. While we have demonstrated that co-culture of THP-1 macrophages and NTP-exposed cells amplified the release of pro-inflammatory cytokines [22], it is unknown how the responses of these macrophages are regulated over time in co-culture with CD8+ T lymphocytes. Additionally, certain cytokines, such as IL-12, are key to building a CD8+ T-lymphocyte response in viral infections [53,54]. In this context, characterization of cytokines emitted by APCs that have engulfed NTP-exposed cells will also be a future priority in order to better understand their likely role in favoring a CD8+ T-lymphocyte response over a CD4+ T-lymphocyte response. Importantly, experiments with primary monocyte-derived APCs from HIV-1-infected individuals, which may have impaired functions relative to HIV-1 infection, will provide validation of the immunostimulatory potential of NTP.

We also evaluated the effect of NTP exposure on the direct effector functions of CD8+ T lymphocytes, which play a crucial role in eliminating infected cells [55] and, thus, need to be considered in an ex vivo approach, whereby NTP-exposed infected cells are readministered to HIV-1 patients. While these CD8 T+ lymphocytes help limit viral spread, immediate clearance of administered NTP-exposed HIV-1-infected cells may also compromise the availability of DAMPs from inoculated cells and impact APC functions that are necessary for building systemic anti-HIV-1 immunity. We showed that NTP did not enhance the antigenicity of J-Lat clones 10.6 and 15.4, as demonstrated by a lack of CD8+ T-lymphocyte proliferation in co-culture with these cells when compared to CD8+ T lymphocytes cultured in the absence of J-Lat cells (Figure 4). This lack of CD8+ T-lymphocyte stimulation would need to be verified using other cell lines and primary cells to rule out NTP-independent factors affecting antigen presentation. Furthermore, if CD8+ T-lymphocyte stimulation is not achieved in a more clinically relevant model, such as HIV-1 latently infected CD4+ T lymphocytes and autologous CD8+ T lymphocytes from patients, this lack of an effect would indicate that there are other factors that contribute to the low immunogenicity of latently infected cells that need to be investigated. Comparisons of CD8+ T lymphocytes responding to latently infected cells and those responding to productively infected cells would help to identify some of these factors. Because numerous reservoirs of latently infected cells exist outside the peripheral blood in ART-suppressed PLWH [56], driving a systemic CD8+ T-lymphocyte response through immunotherapeutic activation of CD8+ T lymphocytes in the lymph nodes is key. Therefore, direct stimulation of CD8+ T lymphocytes would only allow immediate effects in response to NTP-exposed cells that are introduced in vivo. This interaction represents only one possible local response among others with innate immune cells that can serve to control viral spread.

Because our studies to date have relied on cell line models of latently infected cells and APCs, future studies are needed to address several aspects of the proposed therapy that we recognize as challenges to its development. The first is our hypothesis that the NTP-altered epitope presentation will invoke adaptive immune responses to sub-dominant viral epitopes, resulting in broader CD8+ T-lymphocyte cell responses and more effective control of HIV-1 reemergence from latently infected cells. Our previous studies [22] suggest that NTP exposure alters the array of MHC class I-associated peptides presented on latently infected cells. Future studies in a murine model of HIV-1 infection will test this important aspect of the proposed immunotherapy.

The second challenge concerns the number of latently infected cells in cell samples collected from HIV-1-infected patients. In ART-suppressed patients, HIV-1-infected (p24+) cells constitute a very small fraction (<1 to ~100 HIV-1+ cells per 10^6^ total cells) of the total leukocyte population [57]. Furthermore, only a fraction of those HIV-1-positive cells contain replication-competent proviruses [58]. Consequently, leukocytes collected for ex vivo exposure to NTP will likely have insufficient numbers of reservoir T lymphocytes to reach the necessary levels of NTP-induced antigenicity and immunogenicity. In the clinical application of an NTP-based immunotherapy, it may be necessary to activate latently infected reservoir cells and/or promote their proliferation ex vivo to boost viral gene expression before the application of NTP.

A third important challenge is the need to induce viral gene expression in reservoirs to make them visible to adaptive immune responses. For this therapy to succeed as a replacement for continuous ART in PLWH, the NTP-modulated immune response must provide effective control of HIV-1 emerging from viral reservoirs following the cessation (or failure) of ART. Viral latency is a major contributor to the low immunogenicity of HIV-1-infected cells, which comprise the vast majority of infected cells in ART-suppressed patients [59]. Systemic stimulation of viral gene expression in latently infected cells would enable engagement of various antiviral responses, including the release of DAMPs that alert and recruit APCs, and viral antigen presentation as a consequence of stimulated HIV-1 expression and replication [59]. This is a significant challenge in our approach because (i) ex vivo application of NTP will not cause the reactivation of latent reservoirs and (ii) the direct application of NTP to all latently infected cells to stimulate reactivation is clearly not feasible. Therefore, a combinatorial approach involving the administration of an LRA, followed by administration of NTP-exposed patient cells, may be required to develop an effective cure strategy that targets viral reservoirs throughout the body in PLWH. Alternatively, the NTP-based immunotherapy could be preceded by a structured treatment interruption, which would also facilitate systemic virus reactivation targeted by an effective immunotherapy-induced immune response.

Comparisons of the proposed therapy with similar immunotherapy approaches suggest an advantage of an NTP-based regimen. In a phase I trial involving autologous leukocytes pulsed with overlapping Gag peptides [60], the therapeutic regimen was not significantly immunogenic. This lack of efficacy may be speculatively attributed to its focus solely on Gag as the target antigen. In contrast, a study using autologous monocyte-derived dendritic cells pulsed with autologous, heat-inactivated HIV-1 reported increases in HIV-1-specific T-lymphocyte responses and reductions in viral loads after treatment [61]. However, even heat-inactivated virus is limited in its antigenicity because viral particles lack antigens expressed during productive replication (e.g., Rev, Tat, and Nef). Indeed, the successes achieved in vaccination trials using Tat as the target antigen [62] suggest the value of a full range of antigens in an effective therapeutic vaccine. We speculate that an NTP-based immunotherapy will have similar or greater effectiveness relative to approaches based on heat-inactivated virus because antigenicity will be provided by expression of all viral proteins during productive ex vivo HIV-1 replication in autologous reservoir T lymphocytes. Full antigenicity combined with NTP-modified patterns of antigen presentation [22] should facilitate effective immune responses against recrudescent HIV-1 infection.

## 5. Conclusions

These studies reflect progress in the development of an NTP-based immunotherapy for PLWH. An immunotherapy that provides NTP-enhanced immune responses that offer long-term, drug-free control of HIV-1 reemergence from latent infection will be highly beneficial to millions of PLWH around the world.

## Figures and Tables

**Figure 1 biomedicines-11-00122-f001:**
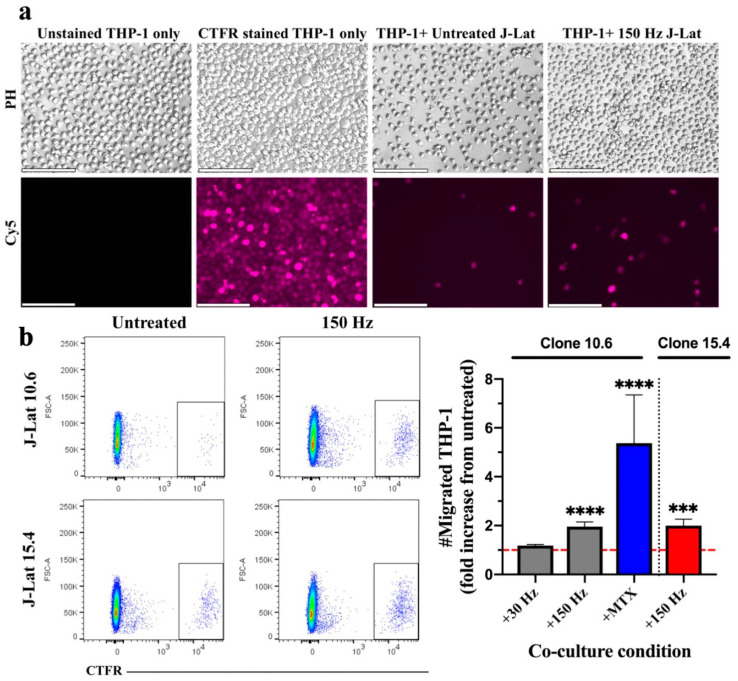
THP-1 monocyte migration is stimulated in response to NTP exposure of J-Lat cells. (**a**) Epifluorescence microscopy images of the basolateral chamber of the transwell system (20x magnification, scale bar is 100 µm), showing CTFR-labeled THP-1 monocytes that migrated across the transwell insert into the basolateral chamber containing J-Lat cells (clone 10.6). Unstained and CTFR-stained THP-1 cells only were cultured in the basolateral chamber as negative and positive controls, respectively, for detection of labeled THP-1 cells. Abbreviations: CTFR, Cell-Trace Far Red; PH, phase contrast; Cy5, Cyanine 5 (used CTFR fluorescence). (**b**) The number of migrated THP-1 monocytes (CTFR-labeled) 24 h later increased in the presence of J-Lat cell clones 10.6 or 15.4 exposed to NTP at 150 Hz, as shown by representative scatter plots and the graph (gray, clone 10.6 exposed to NTP; blue, clone 10.6 exposed to MTX; red, clone 15.4 exposed to NTP). The dotted red line in the graph indicates no change with respect to cells not exposed to NTP (untreated). Mitoxantrone (MTX) was used as positive control for migration. Statistical significance was calculated using the Kruskal–Wallis test, according to the NTP-naïve (untreated) J-Lat co-culture control (*** *p* < 0.001, **** *p* < 0.0001).

**Figure 2 biomedicines-11-00122-f002:**
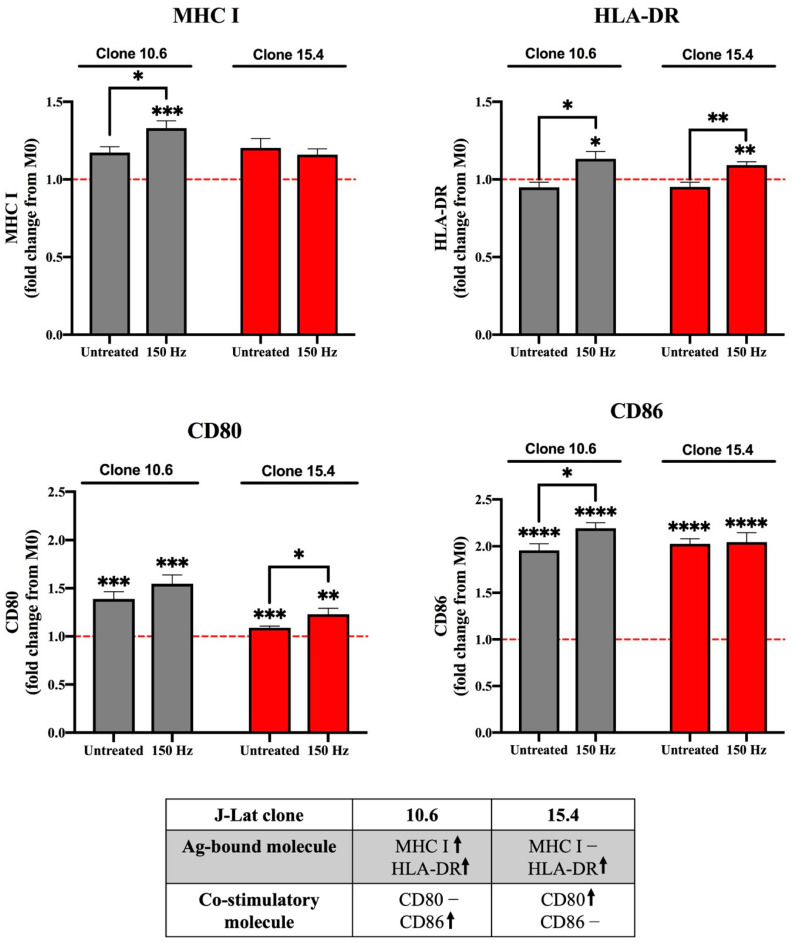
Phagocytic uptake of NTP-exposed J-Lat clones promotes maturation of M0 macrophages. M0 macrophages that engulf NTP-exposed (150 Hz for 10 s) J-Lat cells (gray, clone 10.6; red, clone 15.4) have altered expression of maturation markers MHC I, HLA-DR (MHC II), CD80, and/or CD86. The red dotted line in each graph indicates no change with respect to M0 macrophages cultured without J-Lat cells. The table summarizes changes in expression in the antigen (Ag)-presenting molecule or co-stimulatory molecule on macrophages that engulfed NTP-exposed J-Lat cells relative to M0 macrophages cultured alone (**↑** and – indicate an increase and no change, respectively). Data are presented as mean ± SEM from four independent experiments, with three replicates per condition in each of the three experiments. Statistical significance was calculated using a Brown–Forsythe and Welch’s Anova test, by comparing the mean fluorescence intensity (MFI) of MHC I, HLA-DR, CD80, or CD86 on matured macrophages to M0 macrophages cultured in the absence of J-Lat cells (* *p* ≤ 0.05, ** *p* < 0.01 *** *p* < 0.001, **** *p* < 0.0001).

**Figure 3 biomedicines-11-00122-f003:**
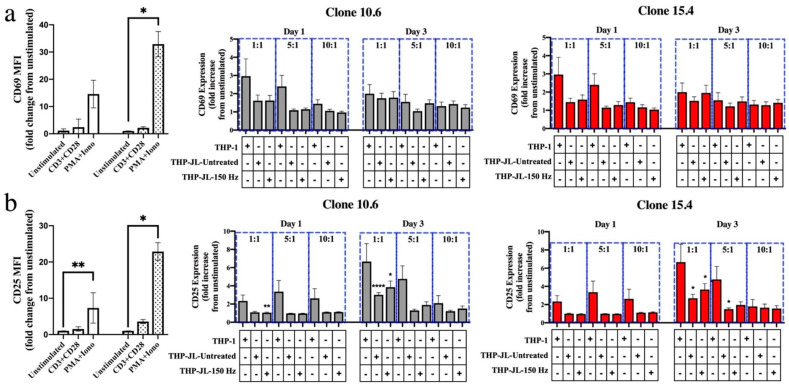
NTP does not stimulate efficient cross-presentation of J-Lat cell-derived antigens on THP-1 macrophages. (**a**) The surface presentation of activation markers CD69 or (**b**) CD25 on CD8+ T lymphocytes in co-culture with M0 macrophages alone (THP-1) or those that had engulfed NTP-naïve (THP-JL-Untreated) or NTP-exposed cells (THP-JL-150 Hz). The fold change in surface display of activation markers for all experimental conditions was calculated relative to the unstimulated CD8+ T-lymphocyte control cultured in the absence of THP-1 macrophages. CD8+ T lymphocytes treated with an anti-CD3 and anti-CD28 antibody cocktail or a PMA and ionomycin cocktail (PMA + Iono) served as positive controls for activation. Cell activation was assessed at days 1 and 3 after introduction of the CD8+ T lymphocytes. Gray and red bars indicate J-Lat clones 10.6 and 15.4, respectively. Data are presented as mean ± SEM from three independent experiments (*n* = 3 donors), with three replicates per condition. Statistical significance was calculated relative to the unstimulated CD8+ T-lymphocyte control, as well as the CD8+ T lymphocyte and THP-1 macrophage negative control using the Brown–Forsythe and Welch’s Anova tests with Dunnett’s post hoc test. (* *p* ≤ 0.05, ** *p* < 0.01, **** *p* < 0.0001).

**Figure 4 biomedicines-11-00122-f004:**
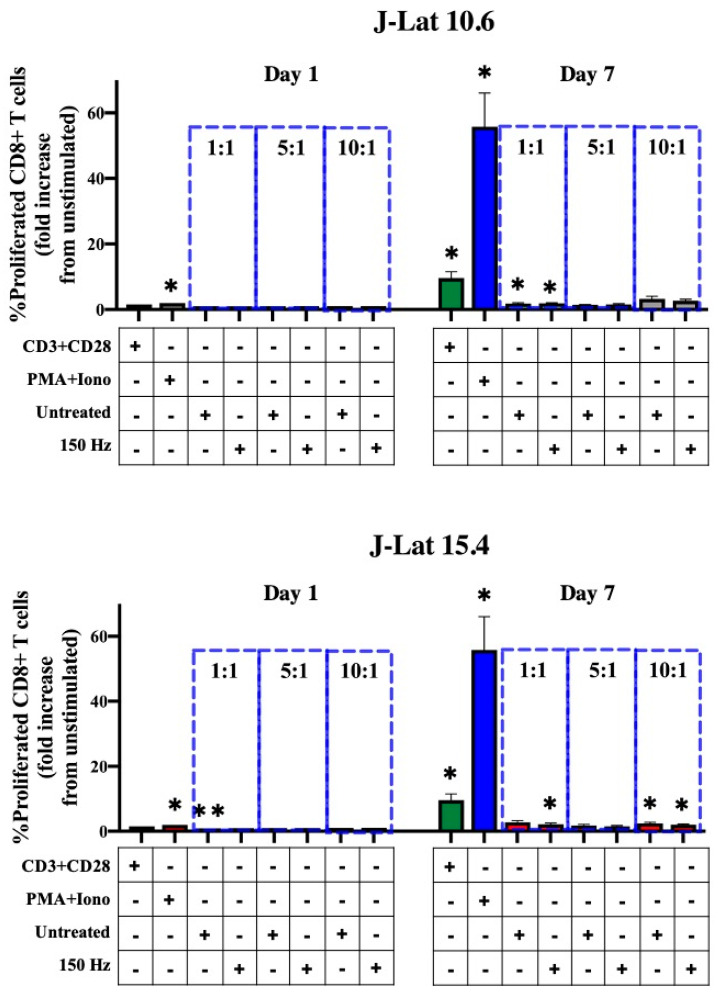
CD8+ T-lymphocyte proliferation is not stimulated in co-culture with NTP-exposed J-Lat cells. Summary of the percent proliferated CD8+ T lymphocytes (CTFRlo) demonstrating no statistically significant increase in proliferated CD8+ T lymphocytes co-cultured with NTP-exposed cells compared to those co-cultured with untreated J-Lat controls at any of the effector:target (E:T) ratios (1:1, 5:1, or 10:1). Gray and red bars indicate J-Lat clones 10.6 and 15.4, respectively. CD8+ T lymphocytes treated with an anti-CD3 and anti-CD28 antibody cocktail (CD3 + CD28; green bars) or a PMA and ionomycin cocktail (PMA + Iono; blue bars) served as positive controls for proliferation. Data are presented as mean ± SEM from three independent experiments (*n* = 3 donors), with three replicates per condition. Statistical significance was calculated according to the unstimulated CD8+ T-lymphocyte control using the one-way Anova with Dunnett’s post-hoc test (* *p* ≤ 0.05, ** *p* < 0.01).

**Figure 5 biomedicines-11-00122-f005:**
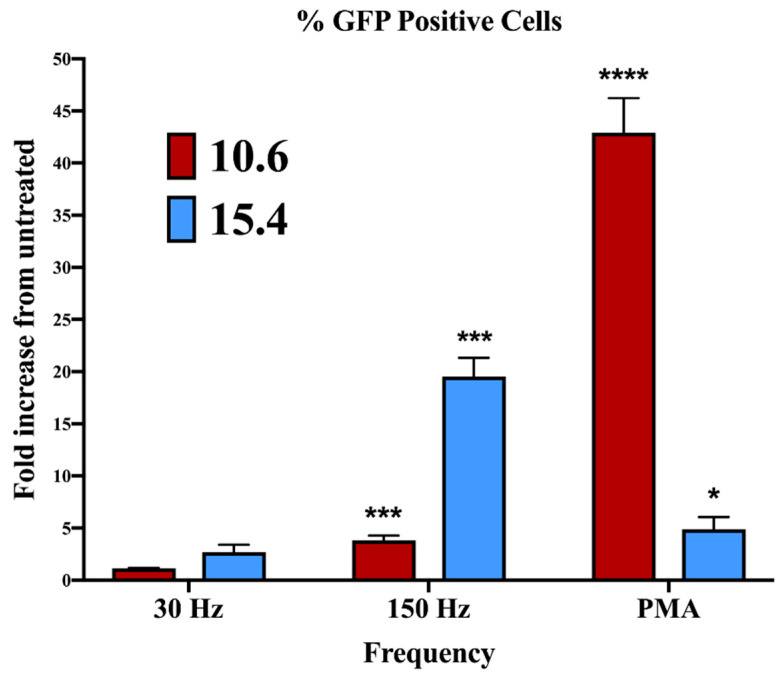
The magnitude of NTP-mediated stimulation of HIV-1 gene expression in J-Lat cells differs between clones. J-Lat cells were exposed to 30 or 150 Hz NTP or 100 nM PMA for 24 h prior to assessing the percentage of GFP-expressing cells by flow cytometry. Data are presented as mean ± SEM from three independent experiments. Statistical significance was calculated using a one-way Anova with Dunnett’s post-hoc test, comparing the percent of GFP-positive cells for each clone with their respective untreated control (* *p* < 0.05, *** *p* < 0.001, **** *p* < 0.0001).

## Data Availability

The data presented in this study are available on request from the corresponding author. The data are recorded locally and are not publicly available.

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
