# Peer review of "Immunomodulatory Effects of Non-Thermal Plasma in a Model for Latent HIV-1 Infection: Implications for an HIV-1-Specific Immunotherapy"

_biomedicines, 2023, doi:10.3390/biomedicines11010122_

Round 1

Reviewer 1 Report

The manuscript reports an interesting study and fits well with the journal's scope. I have only one major concern and some minor points to be addressed:
I think the macrophage maturation and CD8+ cell activation should be further analyzed by quantifying cytokines/chemokines, this would make the study a robust piece of research.
Minor points:
The text in lines 90-101 and figure 6 should be deleted, this is the aim of the group, not the objective of this study.
The main flaw in the methodology is the lack of details on flow cytometry parameters. Please include them in the corresponding sections.
Figure 1A, images require scale bars.

Reviewer 2 Report

The manuscript by Mohamed et al., entitled: Immunomodulatory effects of non-thermal plasma in a model for latent HIV-1 infection: Implications for immunotherapy against HIV-1 infection, is exciting and well-planned. The study documents the development of NTP-based immunotherapy for latent HIV infections. Overall, the study is impactful, but there are a few concerns. Please find my comments below that might help to improve the quality of the manuscript. 

A few places are missing references. For example, the Discussion section, paragraph#2. Additionally, adding a few lines about NTP and its use for other diseases (apart from cancer, which is discussed) will be helpful. 

The authors should briefly discuss current targeted latency reactivation strategies and their drawbacks.

The authors should include cell viability data upon NTP treatment. 

Figure 5 shows a significant clone-specific response. Any possible explanation for why this is happening? Additionally, It will be great to validate this in latently infected primary cells by measuring the viral load. 

HIV infection and HIV latency reactivation are different and should not be used interchangeably. The study is performed in J-Lats and does not involve any infections (Only reactivation). Hence statements like Line #649, "drug-free control of HIV-1 infection," should be corrected to "drug-free control of latent HIV-1 infection" throughout the manuscript.

Add catalog numbers in the methods sections of all the antibodies used.  

Round 2

Reviewer 1 Report

Thanks for addressing my concerns. However, I still think that figure 6
should be deleted, this is the aim of the group, not the objective of this
study. It can mislead the readers.
